# Risk Group Stratification for Recurrence-Free Survival and Early Tumor Recurrence after Radiofrequency Ablation for Hepatocellular Carcinoma

**DOI:** 10.3390/cancers15030687

**Published:** 2023-01-22

**Authors:** Dong Ik Cha, Soo Hyun Ahn, Min Woo Lee, Woo Kyoung Jeong, Kyoung Doo Song, Tae Wook Kang, Hyunchul Rhim

**Affiliations:** 1Department of Radiology, Samsung Medical Center, Sungkyunkwan University School of Medicine, 81 Irwon-ro, Gangnam-gu, Seoul 06351, Republic of Korea; 2Department of Mathematics, Ajou University, Suwon 16499, Republic of Korea; 3Samsung Advanced Institute for Health Sciences & Technology, Sungkyunkwan University, Seoul 06355, Republic of Korea

**Keywords:** predictive model, nomogram, radiofrequency ablation, carcinoma, hepatocellular, prognosis, survival times

## Abstract

**Simple Summary:**

Early detection and treatment of recurrence after radiofrequency ablation (RFA) for hepatocellular carcinoma (HCC) are important steps to improving patient survival. Although the prognosis after RFA for HCC may vary according to different risk levels, there is no standardized follow-up protocol according to each patient’s risk. This study aimed to stratify the patients according to their risk of recurrence-free survival (RFS) and early (≤2 years) tumor recurrence (ETR) based on predictive models and to evaluate whether the risk groups show differences in restricted mean survival times after RFA for HCC. Our predictive models were able to stratify patients into three groups according to their risk of RFS and ETR. The risk groups showed differences in RMSTs, which may be used to establish different follow-up protocols for the three risk groups.

**Abstract:**

Purpose: Although the prognosis after radiofrequency ablation (RFA) for hepatocellular carcinoma (HCC) may vary according to different risk levels, there is no standardized follow-up protocol according to each patient’s risk. This study aimed to stratify patients according to their risk of recurrence-free survival (RFS) and early (≤2 years) tumor recurrence (ETR) after RFA for HCC based on predictive models and nomograms and to compare the survival times of the risk groups derived from the models. Methods: Patients who underwent RFA for a single HCC (≤3 cm) between January 2012 and March 2014 (*n* = 152) were retrospectively reviewed. Patients were classified into low-, intermediate-, and high-risk groups based on the total nomogram points for RFS and ETR, respectively, and compared for each outcome. Restricted mean survival times (RMSTs) in the three risk groups were evaluated for both RFS and ETR to quantitatively evaluate the difference in survival times. Results: Predictive models for RFS and ETR were constructed with c-indices of 0.704 and 0.730, respectively. The high- and intermediate-risk groups for RFS had an 8.5-fold and 2.9-fold higher risk of events than the low-risk group (both *p* < 0.001), respectively. The high- and intermediate-risk groups for ETR had a 17.7-fold and 7.0-fold higher risk than the low-risk group (both *p* < 0.001), respectively. The RMST in the high-risk group was significantly lower than that in the other two groups 9 months after RFA, and that in the intermediate-risk group became lower than that in the low-risk group after 21 months with RFS and 24 months with ETR. Conclusion: Our predictive models were able to stratify patients into three groups according to their risk of RFS and ETR after RFA for HCC. Differences in RMSTs may be used to establish different follow-up protocols for the three risk groups.

## 1. Introduction

Radiofrequency ablation (RFA) has been widely used for the treatment of hepatocellular carcinoma (HCC) and is recommended as the first-line treatment option for early-stage HCC not suitable for surgery [1,2]. However, tumor recurrence after ablation for HCC has been reported to be up to 75% at 5 years [3,4,5,6], and negatively affects patient survival [7]. Early detection and treatment of recurrence are important to improving patient survival, but too frequent surveillance can lead to inefficient operation of valuable medical resources and can be stressful to the patients. Therefore, it is important to organize personalized post-procedural follow-up by identifying patients at higher risk of recurrence who require intense surveillance and those at lower risk who may require less intense follow-up.

Several staging or scoring systems have been proposed to predict survival and provide treatment strategies according to tumor stage [8,9]. In addition, extensive efforts have been made to identify the risk factors associated with tumor recurrence and survival in laboratory and imaging studies [7,10,11,12,13,14,15,16]. Various predictive models have been proposed using risk factors to predict post-procedural outcomes in patients with HCC treated with RFA [17,18]. However, new factors associated with tumor recurrence and survival, such as hepatic function markers [albumin-bilirubin (ALBI) grade [19], aspartate aminotransferase (AST)/platelet ratio index (APRI) [20]], tumor location (subcapsular or peri-vessel location) [7], and tumor marker-based and imaging finding-based models for microvascular invasion (MVI) [14,15,16] are being reported. Therefore, an updated predictive model that utilizes all known risk factors to achieve the highest predictability is required.

In this study, we aimed to stratify patients according to their risk of recurrence-free survival (RFS) and early (≤2 years) tumor recurrence after RFA for HCC by developing predictive models and building nomograms that comprehensively include updated laboratory and imaging factors, and to compare the survival times of the risk groups derived from the models for individualized estimation of RFS and early tumor recurrence after RFA for HCC.

## 2. Materials and Methods

### 2.1. Patients

This retrospective study was conducted at a single tertiary academic center, Samsung Medical Center, Sungkyunkwan University, Seoul, Korea. The institutional review board approved this study and waived the requirement for informed consent (IRB No. SMC 2021-11-133). The inclusion criteria for this study were as follows: (a) clinical patients at high risk for HCC, such as those with chronic hepatitis B and liver cirrhosis; (b) a single nodular tumor (<3 cm in size) that was clinically identified as HCC at the time of RFA; (c) a tumor treated using percutaneous RFA between January 2012 and March 2014; (d) no previous treatment history for HCC; (e) no evidence of macrovascular invasion or extrahepatic metastasis (EM) on pretreatment imaging; (f) Child-Pugh class A or B liver function; (g) no other previous or concomitant malignancies; and (h) gadoxetic acid-enhanced liver magnetic resonance imaging (MRI) with diffusion-weighted imaging (DWI) and hepatobiliary phase (HBP) images within 2 months prior to RFA. Patients with inappropriate MRI quality were excluded. Inappropriate MRI quality included missing HBP, severe respiratory motion artifact on arterial phase, too early arterial scanning defined as the absence of contrast agent in portal veins on arterial phase images, and poor quality of DWI.

### 2.2. Clinical and Laboratory Factors

Patient age, sex, cause of liver disease, ALBI grade, APRI, Child-Pugh classification, serum alpha-fetoprotein (AFP), and serum protein induced by vitamin K absence-II (PIVKA-II) levels were assessed. The “Model for tumor recurrence after living donor liver transplantation” (MoRAL) score, a serum tumor marker-based scoring system, was calculated based on serum AFP and PIVKA-II levels, with a cutoff value of 68 to classify lesions at high risk of recurrence [15].

### 2.3. Image Analysis

Two abdominal radiologists (M.W.L. and D.I.C., with 16 and 6 years of experience in liver MRI interpretation, respectively) independently reviewed the images of the hepatic tumors before RFA. Disagreements were resolved by a third reviewer (W.K.J., with 16 years of liver MRI interpretation experience). The reviewed imaging findings included tumor size, imaging features of the Liver Imaging Reporting and Data System (LI-RADS), including non-rim arterial phase hyperenhancement (APHE), non-rim washout appearance, enhancing capsule, LI-RADS category M (LR-M) features [21], peri-vascular location [22], subcapsular location [23], peritumoral parenchymal enhancement in the arterial phase, tumor contour, peritumoral hypointensity on HBP [11], and signal intensity (SI) of the lesion on the HBP [24]. A scoring system to classify HCCs with a high risk of MVI (MVI-high risk) using AFP, PIVKA-II, peritumoral parenchymal enhancement, and peritumoral hypointensity on HBP was calculated [14]. Details of the models used in this study are provided in the Appendix A.

Tumor characteristics such as tumor size and location were determined by one radiologist (M.W.L., with 16 years of experience in liver MRI interpretation and tumor ablation). Tumor size was measured at the best visible sequence. The location of the tumor was determined according to its relationship with the liver capsule (subcapsular or non-subcapsular) and intrahepatic vessels (portal and hepatic vein) with a diameter of 3 mm or larger. A subcapsular tumor was defined as one that was located within 0.1 cm of the liver capsule [23].

### 2.4. RFA Procedure and Follow-Up Protocol after Treatment

RFA was performed on an inpatient basis by one of five radiologists (H.K.L., H.R., M.W.L., T.W.K., or K.D.S.) with >3 years of experience in locoregional treatments for hepatic tumors. A multiphase liver CT was performed immediately after RFA to evaluate technical success. Multiphase liver computed tomography (CT) and laboratory tests, including tumor markers, were performed 1 month after discharge, followed by every 3 months for the first 2 years, and every 4–6 months thereafter [25].

### 2.5. Outcome Assessment

The primary outcomes were RFS and early tumor recurrence after RFA. RFS was defined as the development of recurrence or death after RFA. Recurrence included local tumor progression (LTP) in the ablation zone, intrahepatic distant recurrence (IDR), and EM. LTP was defined as the appearance of the foci of disease in tumors during follow-up that were previously considered to be completely ablated [26]. IDR was defined as the development of tumors away from the ablation zone, and EM refers to all tumor lesions diagnosed outside the liver [27].

### 2.6. Patient Risk Stratification According to Predictive Models and Nomograms

Predictive models and nomograms were developed for RFS and early tumor recurrence, respectively. Based on the total nomogram points, the patients were classified into low-, intermediate-, and high-risk groups for RFS or early tumor recurrence. Patients with total nomogram points in the lower quartile (<25%) were classified as low-risk, those in the upper quartile (>25%) as high-risk, and those between (25–75%) were classified as intermediate-risk. Subsequently, the risk of recurrence or death for RFS and the risk of tumor recurrence within 2 years after RFA for early tumor recurrence in the high- and intermediate-risk groups were evaluated and compared with the low-risk group.

Restricted mean survival times (RMSTs) in the low-, intermediate-, and high-risk groups were evaluated for both RFS and early tumor recurrence to quantitatively evaluate the difference in survival times among the three groups during follow-up.

### 2.7. Statistical Analysis

The characteristics of the patients were summarized using means with standard deviations or medians with ranges for continuous data and numbers with percentages for categorical data. Kaplan-Meier (KM) curves for RFS and cumulative incidence rate curves for early tumor recurrence with the log-rank test and a univariable and multivariable Cox proportional hazard regression model were performed for survival analysis. Multivariable analyses using a stepwise variable selection method based on the Akaike information criterion (AIC) were performed to build predictive models [28]. The candidate variables for the variable selection were chosen to avoid multicollinearity. The variance inflation factor was used to measure the severity of multicollinearity among the independent variables included in the model, and a value of less than 10 was deemed acceptable. In specific, ‘MoRAL score > 68′, ‘MVI-high risk group’, and major and LR-M features of the LI-RADS were selected over AFP and PIVKA-II, over AFP, PIVKA-II, peritumoral enhancement, peritumoral hypointensity, and the LI-RADS category, respectively, for multivariable analysis. Statistical significance was set at *p* < 0.05.

RMSTs of the nomogram-based low-, intermediate-, and high-risk patients were evaluated by measuring the area above the cumulative incidence curve for early tumor recurrence from the RFA procedure time to the evaluated time point. The RMSTs of the three groups were compared.

The performance of predictive models was evaluated using the concordance index (c-index) with 95% confidence intervals (CI). The predictive models were internally validated using the bootstrap resampling method with 1000 replicates. The performance of the bootstrapped sample applied to the predictive models was estimated, and the model was determined to be valid if the bootstrapped sample showed a similar level of performance.

To evaluate inter-reader agreement based on independent image review, weighted kappa statistics with 95% CIs were used. All imaging variables had a substantial or excellent degree of agreement, and detailed results are provided in the Appendix A. Statistical analysis was performed using R version 3.5.0 (The R Foundation for Statistical Computing, Vienna, Austria).

## 3. Results

One hundred and fifty-two patients were included in this study (Figure 1). Among them, recurrence or death for RFS occurred in 97 patients, and early tumor recurrence occurred in 54 patients. Descriptive data are presented in Table 1.

### 3.1. Predictive Model for RFS

The RFS rates at 1, 2, and 5 years were 83.9% (95% CI 78.2–90.0%), 61.5% (95% CI 54.1–69.9%), and 37.6% (95% CI 30.4–46.5%), respectively. Table 2 shows the results of the univariable and multivariable analyses for RFS. On multivariable analyses, age, ALBI grade 2 (reference: grade 1), APRI, MoRAL score > 68, subcapsular location, non-rim hyperenhancement (reference: no APHE), enhancing capsule (reference: no enhancing capsule), and MVI-high risk were found to be associated with RFS. To build a predictive model with the highest predictive performance, Child-Pugh classification B (reference: A) and the SI of the lesion on HBP (low SI, reference: iso/high) were also included in the predictive model.

### 3.2. Patient Risk Stratification for RFS and Their Comparisons

A nomogram was constructed using the results of the multivariable analysis (Figure 2). The lower quartile was 185.75 points and the upper quartile was 242.56 points; patients were classified into low-, intermediate-, and high-risk groups according to their points. Patients in the high-risk group had an 8.5-fold higher risk of recurrence or death than those in the low-risk group (95% CI 4.3–16.9, *p* < 0.001), and those in the intermediate-risk group had a 2.9-fold higher risk than those in the low-risk group (95% CI 1.6–5.5, *p* < 0.001), with a c-index of 0.675 (95% CI 0.626–0.724). RFS rates were significantly different among the three groups (Figure 2C, *p* < 0.001).

The RMSTs of the three risk groups are shown in Table 3 and in Figure 2C. The RMSTs of the three groups were not significantly different until 6 months. However, at 9 months, the RMST in the high-risk group was 8.192 months (95% CI 7.610–8.774 months); in other words, the restricted mean time loss (RMTL) until 9 months was 0.808 months, which was significantly different from the low-risk group (RMST 8.858 [95% CI 8.583–9.133 months], or RMTL 0.142, *p* = 0.043) and the intermediate-risk group (RMST 8.877 [95% CI 8.736–9.018 months], or RMTL 0.123, *p* = 0.025). The RMST of the high-risk group was significantly lower than that of the other groups and was sustained until the last time point, which was 60 months. A significant difference between the intermediate- and low-risk groups occurred at 21 months after RFA (intermediate-risk group, RMST 18.549 months [95% CI 17.555–19.543 months]; and low-risk group, RMST 20.116 months [95% CI 19.056–21.177 months]; *p* = 0.035).

### 3.3. Predictive Model for Early Tumor Recurrence

The cumulative incidence rates of early tumor recurrence at 1 and 2 years were 15.1% (95% CI 9.2–20.6%) and 35.5% (95% CI 27.5–42.7%), respectively. Table 4 shows the results of the univariable and multivariable analyses to predict early tumor recurrence within 2 years after RFA. On multivariable analyses, age, ALBI grade 2, MoRAL score > 68, non-rim hyperenhancement, and MVI-high risk were found to be associated with early tumor recurrence. Enhancing capsule and SI of the lesion on HBP (low SI, reference = iso/high) were included in the predictive model to achieve the highest performance.

### 3.4. Patient Risk Stratification for Early Tumor Recurrence and Their Comparisons

A nomogram was constructed (Figure 3). The lower quartile was 210.80 points and the upper quartile was 289.66 points; patients were classified into low-, intermediate-, and high-risk groups according to their total nomogram points. Patients in the high-risk group had a 17.7-fold higher risk of early tumor recurrence than those in the low-risk group (95% CI 4.184–75.200, *p* < 0.001), and those in the intermediate-risk group had a 7.0-fold higher risk than those in the low-risk group (95% CI 1.665–29.360, *p* < 0.001), with a c-index of 0.696 (95% CI 0.633–0.759). Early tumor recurrence rates differed significantly among the three groups (Figure 3C, *p* < 0.001).

The RMSTs of the three risk groups are presented in Table 5 and in Figure 3C. Similar to the RFS, the RMSTs were not significantly different until 6 months. At 9 months, the RMST in the high-risk group, with an RMST of 8.203 months (95% CI 7.647–8.758 months), was significantly lower than that in other groups (low-risk group RMST 8.846 [95% CI 8.548–9.144 months], *p* = 0.045; and intermediate-risk group RMST 8.929 [95% CI 8.803–9.055 months], *p* = 0.012]). A significant difference between the intermediate- and low-risk groups occurred at 24 months after RFA (intermediate-risk group, RMST 21.077 months [95% CI 19.987–22.167 months]; and low-risk group, RMST 23.029 months [95% CI 21.694–24.363 months]; *p* = 0.026]).

### 3.5. Diagnostic Performance of the Predictive Models

The c-index of the predictive model for RFS was 0.704 (95% CI: 0.624–0.757). The c-index of the bootstrap sample was 0.717 (95% CI: 0.664–0.770), suggesting that the predictive model for RFS was valid.

The c-index of the predictive model for early tumor recurrence was 0.730 (95% CI: 0.663–0.797), and that of the bootstrap sample was 0.746 (95% CI: 0.678–0.813), indicating that this model was also valid.

## 4. Discussion

In this study, two predictive models for RFS and early tumor recurrence after percutaneous RFA for HCC were proposed. The two models showed high performance in predicting their outcomes, and internal validation showed that the two models were valid. A nomogram was constructed for individualized estimation of the outcomes to be used in daily practice. The patients were divided into low-, intermediate-, and high-risk groups based on their nomogram points, and the three groups showed a significant difference in RFS and early tumor recurrence. Likewise, RMSTs during follow-up were significantly different among the three risk groups for both RFS and early tumor recurrence.

The RMST is a measure that can be interpreted as the average event-free survival time up to a pre-specified, clinically important timepoint [29]. It is useful when proportional hazards cannot be assumed or when event rate is low. In our study, the differences in RMSTs may be utilized as a yardstick to establish different follow-up protocols for the three risk groups. The previous follow-up protocol in our institution was every 3 months for the first 2 years and every 4–6 months thereafter. However, according to our results, the high-risk group seems to need more persistent and intense follow-up than other groups, even if no recurrence occurred during the first 2 years, as the RMST was consistently lower. Meanwhile, the low- and intermediate-risk groups showed similar RMSTs until 18–21 months, possibly indicating that they may require a similar follow-up protocol after RFA. However, in contrast to our routine practice of lengthening the follow-up interval, if there is no recurrence, our study shows that the intermediate-risk group needs more intense follow-up than the low-risk group 21–24 months after RFA. This may be explained by our predictive models having liver function variables, and patients in the intermediate-risk group may have had higher nomogram scores due to the liver function variables. As liver function affects de novo late tumor recurrence, usually 24 months after treatment, liver function may cause differences in RMSTs between the low- and intermediate-risk groups 18–21 months after RFA. Although the optimal follow-up protocol according to patient risk should be evaluated further by multicenter prospective studies, a new follow-up protocol after treatment according to their risk can be carefully proposed. The high-risk group seems to need intense follow-up after treatment, such as every 3 months. Intermediate and low-risk groups need less intensive follow-up, such as 4–6 months, until 18–21 months after treatment. While it seems that the low-risk group can lengthen the follow-up interval to 6–9 months after 21 months, the intermediate group may need to keep the follow-up interval of 4–6 months after 21 months as the risk of recurrence is higher than the low-risk group. A comparison of the median survival time was not possible due to an insufficient number of events in the low-risk group.

Variables used in the predictive model for RFS were the patient’s age, ALBI grade, APRI grade, Child-Pugh grade, MoRAL score > 68, subcapsular location of the tumor, non-rim APHE, enhancing capsule, the low signal intensity of the tumor on HBP on gadoxetic acid-enhanced MRI, and being positive for high risk by the MVI model. Variables for early tumor recurrence were similar but without APRI, Child-Pugh grade, and subcapsular tumor location.

ALBI grade, APRI grade, and Child-Pugh grade are indices of liver function. While the Child-Pugh grade is a traditional system representing liver function, the ALBI grade is an objective liver function index that does not require subjective variables such as ascites and encephalopathy and independently influences survival in patients with HCC [30]. This previous study showed that ALBI revealed two classes with clearly different prognoses among HCC patients with Child-Pugh grade A. This may explain why both Child-Pugh grade and ALBI grade were selected independently without multicollinearity for RFS in our study.

Imaging variables may have inter-reader variability [31,32], which may be an obstacle to using such variables. However, the inter-reader agreements for all imaging variables were substantial or excellent in our study. We believe that inter-reader variability can be minimized with proper training and the establishment of clear standards for imaging findings.

Our study has several limitations. First, this was a retrospective study conducted at a single medical center, which would have led to some degree of selection bias. Second, this study was performed on the hepatitis B virus (HBV)-predominant population. Therefore, the results of our study may not be generalizable to populations where HBV is not the dominant cause of liver disease. Third, external validation was not performed to verify the results. However, internal validation using the bootstrap resampling method was performed, which is a widely used and accurate method for internal validation, and showed that our model was valid. Fourth, a pathologic diagnosis of hepatic lesions that underwent ablation was not made. This was inevitable because it was not routine to perform a biopsy and pathologically diagnose the lesions before performing ablation. However, our cohort is highly biased towards HCC since hepatic tumors suspected to be non-HCC (based on imaging findings or elevated CA19-9) would not have undergone RFA in the first place since RFA is not attempted in patients with tumors with strong features of non-HCC malignancy on pretreatment CT or MRI. Furthermore, the fact that LR-M was not a significant factor affecting the outcomes may suggest that this would not significantly affect the main outcomes of our study.

## 5. Conclusions

In conclusion, our predictive models were able to stratify patients into three groups according to their risk of RFS and early tumor recurrence after RFA for HCC. Differences in the RMST may be used to establish a different follow-up protocol for the three risk groups.

## Figures and Tables

**Figure 1 cancers-15-00687-f001:**
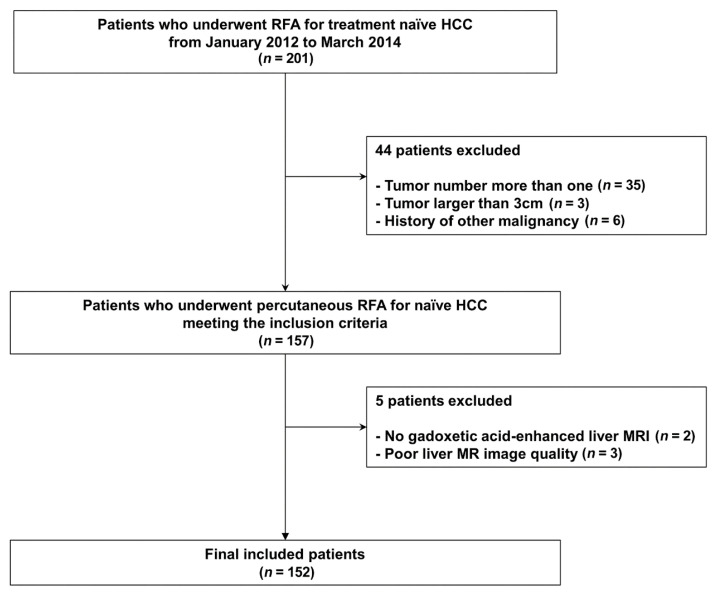
Patient inclusion process.

**Figure 2 cancers-15-00687-f002:**
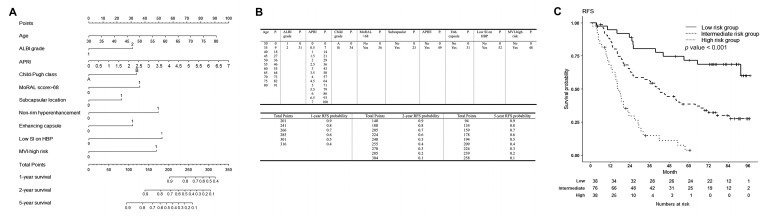
Nomogram to estimate recurrence-free survival (RFS) after percutaneous radiofrequency ablation for HCC, and nomogram-based low-, intermediate- and high-risk groups for RFS. (**A**) The included variables in the nomogram for RFS are age, Albumin-Bilirubin grade 2 (reference: grade 1), aspartate aminotransferase/platelet ratio index, Child-Pugh classification B (reference A), MoRAL score > 68, subcapsular location, non-rim hyperenhancement of the lesion (reference: no enhancement), enhancing capsule (reference: no enhancing capsule), low signal intensity (SI) of the lesion on hepatobiliary phase on gadoxetic acid-enhanced liver MRI (reference: iso or high SI), and microvascular invasion-high risk group. (**B**) The prognostic points of each variable are shown in the upper table, and the estimated RFS rate at 1, 2, and 5 years after RFS for HCC according to the total points are shown in the lower table. (**C**) For RFS, patients with nomogram scores in the lower quartile (<25%, 185.75 points) were classified as low-risk, those in the upper quartile (>25%, 242.56 points) as high-risk, and those in between (25–75%) were classified as intermediate-risk groups. The RFS rates of the low, intermediate, and high risk groups were 94.6% (95% CI 87.6–100%), 86.7% (95% CI 79.3–94.7%), and 68.0% (95% CI 54.6–84.7%) at 1 year; 89.0% (95% CI 79.4–99.8%), 63.9% (95% CI 53.9–75.8%), and 29.5% (95% CI 17.8–48.8%) at 2 years; and 71.7% (95% CI 58.2–88.2%), 38.5% (95% CI 28.7–51.6%), and 3.7% (95% CI 0.6–24%) at 5 years, respectively. The RFS rates among the three groups were significantly different (*p* < 0.001). Patients in the high-risk group had an 8.5-fold higher risk of recurrence or death than the low-risk group (95% CI 4.3–16.9, *p* < 0.001), and those in the intermediate-risk group had a 2.9-fold higher risk than the low-risk group (95% CI 1.6–5.5, *p* < 0.001). At 9 months, the restricted mean survival time (RMST) in the high-risk group was 8.192 months (95% CI 7.610–8.774 months), which was significantly different from the low-risk group (RMST 8.858 [95% CI 8.583–9.133 months], and the intermediate-risk group (RMST 8.877 [95% CI 8.736–9.018 months]. The RMST of the high-risk group was significantly lower than that of the other groups and was sustained until the last time point, which was 60 months. A significant difference between the intermediate- and low-risk groups occurred at 21 months after RFA (intermediate-risk group, RMST 18.549 months [95% CI 17.555–19.543 months]; and low-risk group, RMST 20.116 months [95% CI 19.056–21.177 months]; *p* = 0.035). RFS = recurrence-free survival, HCC = hepatocellular carcinoma, MoRAL = Model for tumor recurrence after living donor liver transplantation, SI = signal intensity, MRI = magnetic resonance imaging.

**Figure 3 cancers-15-00687-f003:**
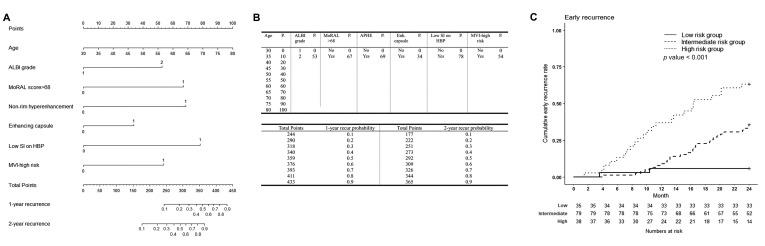
Nomogram to estimate early tumor recurrence after percutaneous radiofrequency ablation for HCC, and nomogram-based low-, intermediate- and high-risk groups for early tumor recurrence. (**A**) The included variables in the nomogram for early tumor recurrence are age, Albumin-Bilirubin grade 2 (reference: grade 1), MoRAL score > 68, non-rim hyperenhancement (reference: no enhancement), enhancing capsule (reference: no enhancing capsule), low signal intensity (SI) of the lesion on hepatobiliary phase on gadoxetic acid-enhanced liver MRI (reference: iso or high SI), and microvascular invasion-high risk group. (**B**) The prognostic points of each variable are shown in the upper table, and the estimated early tumor recurrence rate at 1 and 2 years after RFS for HCC according to the total points are shown in the lower table. (**C**) For early tumor recurrence, patients with nomogram scores in the lower quartile (<25%, 210.80 points) were classified as low-risk, those in the upper quartile (>25%, 289.66 points) as high-risk, and those in between (25–75%) were classified as intermediate-risk groups. The cumulative early tumor recurrence rates of the low-, intermediate-, and high-risk groups were 5.7% (95% CI 0.0–13.1%), 8.9% (95% CI 2.4–14.9%), and 39.5% (95% CI 21.8–53.2%) at 1 year, and 5.7% (95% CI 0.0–13.1%), 35.4% (95% CI 24–45.2%), and 63.2% (95% CI 44.1–75.7%) at 2 years. The cumulative early tumor recurrence rates among the three groups were significantly different (*p* < 0.001). Patients in the high-risk group had a 17.7-fold higher risk of recurrence or death than the low-risk group (95% CI 4.184–75.200, *p* < 0.001), and those in the intermediate-risk group had a 7.0-fold higher risk of recurrence or death than the low-risk group (95% CI 1.665–29.360, *p* < 0.001). At 9 months, restricted mean survival time (RMST) in the high-risk group, with an RMST of 8.203 months (95% CI 7.647–8.758 months), was significantly lower than that in other groups (low-risk group RMST 8.846 [95% CI 8.548–9.144 months], *p* = 0.045; and intermediate-risk group RMST 8.929 [95% CI 8.803–9.055 months], *p* = 0.012]. A significant difference between the intermediate- and low-risk groups occurred at 24 months after RFA (intermediate-risk group, RMST 21.077 months [95% CI 19.987–22.167 months]; and low-risk group, RMST 23.029 months [95% CI 21.694–24.363 months]; *p* = 0.026]. HCC = hepatocellular carcinoma, MoRAL = Model for tumor recurrence after living donor liver transplantation, SI = signal intensity, MRI = magnetic resonance imaging.

**Table 1 cancers-15-00687-t001:** Patient characteristics.

	RFS		Early Tumor Recurrence	
	No Event (*n =* 55)	Event (*n =* 97)	No Event (*n =* 98)	Early Recur (*n =* 54)
Age (year) *	56 (33–77)	57 (31–78)	55 (33–78)	59.5 (31–77)
Sex (male)	12 (21.8)	23 (23.7)	23 (23.5)	12 (22.2)
Cause of liver disease				
HBV	47 (85.5)	78 (80.4)	81 (82.7)	44 (81.5)
HCV	2 (3.6)	12 (12.4)	9 (9.2)	5 (9.3)
Alcohol	1 (1.8)	1 (1)	1 (1)	1 (1.9)
Others	5 (9.1)	6 (6.2)	7 (7.1)	4 (7.4)
ALBI grade				
1	44 (80)	61 (62.9)	75 (76.5)	30 (55.6)
2	11 (20)	36 (37.1)	23 (23.5)	24 (44.4)
APRI *	0.632 (0.22–3)	0.970 (0.207–6.618)	0.857 (0.207–3.969)	0.962 (0.302–6.618)
Child-Pugh classification				
A	52 (94.5)	84 (86.6)	91 (92.9)	45 (83.3)
B	3 (5.5)	13 (13.4)	7 (7.1)	9 (16.7)
AFP (ng/mL) *	6.6 (1.3–1426.0)	14.1 (1.3–2204.6)	8.65 (1.3–1426.0)	17.9 (1.3–2204.6)
PIVKA-II (mAU/mL) *	20 (9–103)	22 (9–11,078)	20 (9–1200)	25 (11–11078)
MoRAL score > 68	18 (32.7)	49 (50.5)	33 (33.7)	34 (63)
Tumor size (cm) *	1.6 (1–2.7)	1.7 (1–2.9)	1.6 (1–2.9)	1.7 (1–2.6)
Tumor location				
Peri-portal vein	2 (3.6)	5 (5.2)	4 (4.1)	3 (5.6)
Peri-hepatic vein	2 (3.6)	9 (9.3)	7 (7.1)	4 (7.4)
Subcapsular	17 (30.9)	40 (41.2)	37 (37.8)	20 (37)
Non-subcapsular	38 (69.1)	57 (58.8)	61 (62.2)	34 (63)
Non-rim hyperenhancement	41 (74.5)	86 (88.7)	78 (79.6)	49 (90.7)
Washout appearance	29 (52.7)	48 (49.5)	46 (46.9)	31 (57.4)
Enhancing capsule	22 (40)	41 (42.3)	37 (37.8)	26 (48.1)
LR-M	13 (23.6)	18 (18.6)	22 (22.4)	9 (16.7)
LI-RADS category				
3	14 (25.5)	23 (23.7)	26 (26.5)	11 (20.4)
4	6 (10.9)	17 (17.5)	14 (14.3)	9 (16.7)
5	22 (40)	39 (40.2)	36 (36.7)	25 (46.3)
M	13 (23.6)	18 (18.6)	22 (22.4)	9 (16.7)
Peri-tumoral enhancement	14 (25.5)	35 (36.1)	29 (29.6)	20 (37)
Non-smooth margin	22 (40)	32 (33)	38 (38.8)	16 (29.6)
Peritumoral hypointensity	3 (5.5)	9 (9.3)	7 (7.1)	5 (9.3)
Low SI on HBP (reference = iso/high)	50 (90.9)	93 (95.9)	98 (91.8)	53 (98.1)
MVI-high risk group	3 (5.5)	13 (13.4)	8 (8.2)	8 (14.8)

Note.—Unless otherwise indicated, data are the number of patients (lesions) with percentages in parentheses. * Data are medians with ranges in parentheses. HBV = hepatitis B virus, HCV = hepatitis C virus, ALBI grade = albumin-bilirubin grade, APRI = AST/platelet ratio index, AFP = alpha-fetoprotein, PIVKA-II = protein induced by vitamin K absence-II, MoRAL score = ‘model for tumor recurrence after living donor liver transplantation’ score, LR-M = LI-RADS category M, LI-RADS = Liver Imaging Reporting and Data System, SI = signal intensity, HBP = hepatobiliary phase, MVI = microvascular invasion, RFS = recurrence free survival.

**Table 2 cancers-15-00687-t002:** Univariate and multivariate analysis using stepwise variable selection for prediction of recurrence-free survival.

Variables	Univariable Analysis			Multivariable Analysis		
	HR	95% CI	*p* Value	HR	95% CI	*p* Value
Age	1.014	0.993–1.036	0.191	1.035	1.011–1.059	0.003
Male (reference = female)	1.173	0.735–1.874	0.503			
Cause of liver disease (reference = HBV)						
HCV	1.865	1.009–3.448	0.047			
Alcohol	1.097	0.152–7.921	0.927			
Others	1.036	0.451–2.38	0.934			
ALBI grade 2 (ref = 1)	2.026	1.334–3.078	0.001	1.800	1.056–3.069	0.031
APRI	1.274	1.056–1.536	0.011	1.308	1.04–1.646	0.022
Child-Pugh classification B (reference = A)	1.959	1.086–3.536	0.026	1.897	0.932–3.863	0.078
MoRAL score > 68	1.694	1.136–2.525	0.010	1.983	1.315–2.992	0.001
Tumor size	1.549	1.004–2.389	0.048			
Peri-portal vein	1.302	0.529–3.206	0.565			
Peri-hepatic vein	1.379	0.694–2.741	0.359			
Subcapsular (reference = non-subcapsular)	1.306	0.871–1.958	0.196	1.551	1.005–2.391	0.047
Non-rim hyperenhancement	1.815	0.968–3.401	0.063	2.533	1.316–4.872	0.005
Washout appearance	0.953	0.639–1.423	0.815			
Enhancing capsule	1.282	0.856–1.92	0.228	1.801	1.167–2.781	0.008
LR-M features	0.839	0.503–1.401	0.502			
Non-smooth margin	0.81	0.53–1.237	0.329			
Low SI on HBP (reference = iso/high)	1.719	0.631–4.68	0.289	2.672	0.927–7.703	0.069
MVI-high risk group	1.925	1.072–3.459	0.028	2.470	1.287–4.743	0.007

HBV = hepatitis B virus, HCV = hepatitis C virus, ALBI grade = albumin-bilirubin grade, APRI = AST/platelet ratio index, MoRAL score = ‘model for tumor recurrence after living donor liver transplantation’ score, LR-M = LI-RADS category M, LI-RADS = Liver Imaging Reporting and Data System, SI = signal intensity, HBP = hepatobiliary phase, MVI = microvascular invasion, HR = hazard ratio, CI = confidence interval. Note—‘MoRAL score > 68′, ‘MVI-high risk group’, and major and LR-M features of the LI-RADS were selected over alpha-fetoprotein (AFP), protein induced by vitamin K absence-II (PIVKA-II), over AFP, PIVKA-II, peritumoral enhancement and peritumoral hypointensity, and over LI-RADS category, respectively, for multivariable analysis.

**Table 3 cancers-15-00687-t003:** Restricted mean survival time of the low-, intermediate-, and high-risk groups for recurrence-free survival, and their comparisons.

Time (Months)	Overall		Low		Intermediate		High		*p*-Value	High		Intermediate
	RMST	95% CI	RMST	95% CI	RMST	95% CI	RMST	95% CI		Intermediate	Low	Low
3	2.990	2.971–3.009	3.000	-	3.000	-	2.961	2.884–3.037		0.311	0.311	-
6	5.900	5.820–5.981	5.937	5.815–6.059	5.972	5.919–6.026	5.721	5.452–5.990		0.072	0.152	0.602
9	8.699	8.517–8.882	8.858	8.583–9.133	8.877	8.736–9.018	8.192	7.610–8.774		0.025	0.043	0.904
12	11.318	11.008–11.629	11.734	11.301–12.168	11.580	11.278–11.882	10.392	9.467–11.317		0.017	0.010	0.567
15	13.759	13.296–14.221	14.572	13.948–15.197	14.080	13.575–14.586	12.325	11.025–13.625		0.014	0.002	0.231
18	16.015	15.384–16.646	17.362	16.531–18.193	16.402	15.661–17.143	13.93	12.264–15.596		0.008	<0.001	0.091
21	18.079	17.269–18.89	20.116	19.056–21.177	18.549	17.555–19.543	15.156	13.155–17.158		0.003	<0.001	0.035
24	20.025	19.024–21.026	22.865	21.561–24.169	20.567	19.304–21.829	16.173	13.842–18.503		0.001	<0.001	0.013
36	26.67	24.871–28.47	32.678	30.197–35.158	27.640	30.197–35.158	18.797	15.362–22.233		<0.001	<0.001	0.004
48	32.379	29.752–35.006	41.993	38.087–45.9	33.613	30.057–37.169	20.344	15.971–24.717		<0.001	<0.001	0.002
60	37.319	33.877–40.760	50.866	45.423–56.31	38.509	33.852–43.165	21.349	16.221–26.477		<0.001	<0.001	0.001

RMST = restricted mean survival time, CI = confidence interval.

**Table 4 cancers-15-00687-t004:** Univariate and multivariate analysis using stepwise variable selection for prediction of early tumor recurrence.

Variables	Univariable Analysis			Multivariable Analysis		
	HR	95% CI	*p* Value	HR	95% CI	*p* Value
Age	1.02	0.993–1.047	0.152	1.033	1.006–1.062	0.018
Male (reference = female)	0.961	0.506–1.826	0.904			
Cause of liver disease (reference = HBV)						
HCV	0.946	0.375–2.385	0.906			
Alcohol	1.364	0.188–9.911	0.759			
Others	0.994	0.357–2.768	0.992			
ALBI grade 2 (ref = 1)	2.051	1.198–3.511	0.009	2.36	1.369–4.069	0.002
APRI	1.119	0.859–1.459	0.405			
Child-Pugh classification B (reference = A)	1.827	0.893–3.741	0.099			
MoRAL score > 68	2.691	1.548–4.680	<0.001	2.985	1.704–5.229	<0.001
Tumor size	1.393	0.788–2.464	0.254			
Peri-portal vein	1.469	0.458–4.706	0.518			
Peri-hepatic vein	1.050	0.379–2.909	0.924			
Subcapsular (reference = non-subcapsular)	1.022	0.589–1.776	0.937			
Non-rim arterial hyperenhancement	2.205	0.878–5.534	0.092	3.067	1.204–7.811	0.019
Washout appearance	1.46	0.851–2.504	0.169			
Enhancing capsule	1.425	0.835–2.431	0.194	1.738	0.998–3.026	0.051
LR-M features	0.737	0.360–1.508	0.403			
Non-smooth margin	0.743	0.414–1.332	0.319			
Low SI on HBP (reference = iso/high)	4.134	0.571–29.901	0.160	3.607	0.492–26.454	0.207
MVI-high risk group	1.816	0.857–3.850	0.119	2.412	1.109–5.245	0.026

HBV = hepatitis B virus, HCV = hepatitis C virus, ALBI grade = albumin-bilirubin grade, APRI = AST/platelet ratio index, MoRAL score = ‘model for tumor recurrence after living donor liver transplantation’ score, LR-M = LI-RADS category M, LI-RADS = Liver Imaging Reporting and Data System, SI = signal intensity, HBP = hepatobiliary phase, MVI = microvascular invasion, HR = hazard ratio, CI = confidence interval. Note—‘MoRAL score > 68′, ‘MVI-high risk group’, and major and LR-M features of the LI-RADS were selected over alpha-fetoprotein (AFP), protein induced by vitamin K absence-II (PIVKA-II), over AFP, PIVKA-II, peritumoral enhancement and peritumoral hypointensity, and over LI-RADS category, respectively, for multivariable analysis.

**Table 5 cancers-15-00687-t005:** Restricted mean survival time of the low-, intermediate-, and high-risk groups for early tumor recurrence, and their comparisons.

Time (Months)	Overall		Low		Intermediate		High		*p*-Value	High		Intermediate
	RMST	95% CI	RMST	95% CI	RMST	95% CI	RMST	95% CI		Intermediate	Low	Low
3	2.990	2.971–3.009	3.000	-	3.000	-	2.961	2.884–3.037		0.311	0.311	-
6	5.908	5.829–5.987	5.931	5.799–6.064	5.973	5.922–6.025	5.750	5.484–6.016		0.106	0.231	0.563
9	8.728	8.553–8.903	8.846	8.548–9.144	8.929	8.803–9.055	8.203	7.647–8.758		0.012	0.045	0.614
12	11.373	11.076–11.67	11.714	11.245–12.184	11.763	11.530–11.997	10.247	9.344–11.151		0.001	0.005	0.855
15	13.844	13.400–14.288	14.543	13.871–15.214	14.390	13.993–14.787	12.066	10.775–13.356		0.001	0.001	0.701
18	16.136	15.529–16.744	17.371	16.484–18.259	16.810	16.209–17.412	13.597	11.929–15.266		<0.001	<0.001	0.305
21	18.261	17.478–19.045	20.200	19.090–21.31	19.009	18.175–19.843	14.921	12.879–16.963		<0.001	<0.001	0.093
24	20.277	19.306–21.247	23.029	21.694–24.363	21.077	19.987–22.167	16.079	13.67–18.488		<0.001	<0.001	0.026

RMST = restricted mean survival time, CI = confidence interval.

## Data Availability

The data that support the findings of this study are not publicly available due to their containing information that could compromise the privacy of research participants but may be available from the corresponding author (M.W.L.) upon reasonable request.

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
