# Peer review of "Risk Group Stratification for Recurrence-Free Survival and Early Tumor Recurrence after Radiofrequency Ablation for Hepatocellular Carcinoma"

_cancers, 2023, doi:10.3390/cancers15030687_

Round 1

Reviewer 1 Report

It is an excellent study about stratification of the risk for recurrence free survival and early tumor recurrence in patients with hcc undergoing treatment with RFA

The methods of the study  are very well presented and the statistical analysis is adequate and comprehensive.  Limitations of the study are the retrospective analysis of the data, as well as the absence of an external validation. However, it is certainly difficult to recruit an adequate number of RFA-treated patients with HCC and follow up the progression of the disease prospectively, in a single-center basis.

Nonetheless,  the internal validation and the estimation of the restricted mean survival times empower the study. The only comment I have to make is why the authors preferred MoRAL score>68’, LR-M features of the LI-RADS, peritumoral enhancement and peritumoral hypointensity over AFP and PIVKA-II in the multivariate analysis? Please explain.

Furthermore, the induction of such a large number of radiological parameters increase the possibility of inter- and intra-observer variability. Please include your comments about that in the section of discussion 

Author Response

R1.

It is an excellent study about stratification of the risk for recurrence free survival and early tumor recurrence in patients with hcc undergoing treatment with RFA

The methods of the study are very well presented and the statistical analysis is adequate and comprehensive. Limitations of the study are the retrospective analysis of the data, as well as the absence of an external validation. However, it is certainly difficult to recruit an adequate number of RFA-treated patients with HCC and follow up the progression of the disease prospectively, in a single-center basis.

R1-1. Nonetheless, the internal validation and the estimation of the restricted mean survival times empower the study. The only comment I have to make is why the authors preferred MoRAL score>68’, LR-M features of the LI-RADS, peritumoral enhancement and peritumoral hypointensity over AFP and PIVKA-II in the multivariate analysis? Please explain.

  • Thank you for your generous comments. We have briefly mentioned in the 2.7. Statistical analysis section (page 5, line 169~172) that the candidate variables for the variable selection were chosen to avoid the multicollinearity. As you have mentioned, it may be a little confusing. We have added the following to the revised manuscript for better understanding.

In specific, ‘MoRAL score>68’, ‘MVI-high risk group’, and major and LR-M features of the LI-RADS were selected over AFP and PIVKA-II, over AFP, PIVKA-II, peritumoral enhancement, and peritumoral hypointensity, and over LI-RADS category, respectively, for multivariable analysis.

R1-2. Furthermore, the induction of such a large number of radiological parameters increase the possibility of inter- and intra-observer variability. Please include your comments about that in the section of discussion

  • You are very right. Imaging variables do have inter- and intra-observer variability. Although the results of the inter-observer agreement are presented in the supplementary section, it will be discussed further in the discussion.

Imaging variables may have inter-reader variability [31,32], which may be an obstacle in using such variables. However, the inter-reader agreements for all imaging variables were substantial or excellent in our study. We believe that inter-reader variability can be minimized with proper training and the establishment of clear standards of the imaging findings.

Reviewer 2 Report

The study focuses on the risk stratification in early-stage hepatocellular carcinoma after radiofrequency ablation by analyzing recurrence-free survival and early tumor recurrence. This topic is widely debated in relation to the high aggressiveness of HCC, with the need to perform RFA properly to obtain better RFS and less metastasis.

Some changes are required before eventual publication.

-One of the aims of this study is to explore the personalized post-procedural follow-up for HCC patients after RFA, the follow-up protocol according to the result (etc. RMST) of this study should be more specifically discussed in Discussion Part.

-The lack of the specific data of postoperative adjuvant therapies, such as chemotherapy and immunotherapy, may affect the result of RFS and early tumor recurrence. It should be mentioned in limitation of this study.

- Due to the lack of pathological diagnosis, the exclusion of intrahepatic cholangiocarcinoma is need. The CA199 and the Imaging diagnosis should be used for exclusion. In this study, the authors need to further specify the approach for clinical diagnosis of HCC rather than ICC.

-The LRM is used instead of LR5 for analysis, but the LRM is used for malignant rather than HCC. This issue should be clarified in the study.

-In the figure of nomogram, the name of the group should be specified rather than 0 and1.

-The patients were divided to 3 groups according to percentage of score<25, 25-75 and >75. The reason for this group needs to be clarified (1/4 to 4 groups or 1/3 to 3 groups).

Author Response

R2

The study focuses on the risk stratification in early-stage hepatocellular carcinoma after radiofrequency ablation by analyzing recurrence-free survival and early tumor recurrence. This topic is widely debated in relation to the high aggressiveness of HCC, with the need to perform RFA properly to obtain better RFS and less metastasis.

Some changes are required before eventual publication.

R2-1. One of the aims of this study is to explore the personalized post-procedural follow-up for HCC patients after RFA, the follow-up protocol according to the result (etc. RMST) of this study should be more specifically discussed in Discussion Part.

  • Thank you for your comment. We were very careful in proposing a follow-up protocol based on our results, as we did not actually evaluate the incidences of recurrence based on a new follow-up protocol. However, as you have mentioned, we would like to briefly mention a possible follow-up protocol based on our results.

Although the optimal follow-up protocol according to patient risk should be evaluated further by multicenter prospective studies, a new follow-up protocol after treatment according to their risk can be carefully proposed. The high-risk group seems to need intense follow-up after treatment, such as every 3 months. Intermediate and low-risk groups need less intense follow-up, such as 4-6 months, until 18-21 months after treatment. While it seems that the low-risk group can lengthen the follow-up interval to 6-9 months after 21 months, the intermediate group may need to keep the follow-up interval of 4-6 months after 21 months as the risk of recurrence is higher than the low-risk group.

R2-2. The lack of the specific data of postoperative adjuvant therapies, such as chemotherapy and immunotherapy, may affect the result of RFS and early tumor recurrence. It should be mentioned in limitation of this study.

  • It is very right to think that adjuvant therapies such as chemotherapy and immunotherapy will affect patient survival. However, it is not standard to use chemotherapy or immunotherapy following RFA if there is no evidence of a viable tumor. The outcomes of our study were RFS and early tumor recurrence, which means that the patients will be censored if recurrence occurs or a patient expires without evidence of disease. Therefore, the effect of such adjuvant therapies would not affect the results of our study. The adjuvant therapies will definitely affect OS, but that was not our investigated outcome. Therefore, the issue of adjuvant therapies following RFA may be beyond the scope of our study.

R2-3. Due to the lack of pathological diagnosis, the exclusion of intrahepatic cholangiocarcinoma is need. The CA199 and the Imaging diagnosis should be used for exclusion. In this study, the authors need to further specify the approach for clinical diagnosis of HCC rather than ICC.

  • You are very right. In page 3 line 90, we stated that the lesions were clinically identified as HCC at the time of diagnosis. Before the introduction of LI-RADS, a diagnosis of HCC was made if a hepatic lesion showed arterial phase hyperenhancement followed by washout in a patient with chronic hepatitis B and liver cirrhosis. Back then, some imaging findings such as subtle rim enhancement, targetoid appearance(s) on diffusion-weighted images or hepatobiliary phase may have been neglected. However, the LR-M of LI-RADS is to maximize the specificity of LR-5 and does not necessarily mean that LR-M is not an HCC. Furthermore, our cohort which is comprised of patients who underwent RFA is highly biased towards HCC since hepatic tumors suspected to be non-HCC (based on imaging findings or elevated CA19-9) would not have undergone RFA in the first place. RFA is not attempted in patients having tumors with strong ICC features on pretreatment CT or MRI. This is probably why the LR-M feature was not a variable affecting the outcomes in our study.

The following has been added to the revised manuscript as the following:

However, our cohort is highly biased towards HCC since hepatic tumors suspected to be non-HCC (based on imaging findings or elevated CA19-9) would not have undergone RFA in the first place since RFA is not attempted in patients having tumors with strong features of non-HCC malignancy on pretreatment CT or MRI. Furthermore, the fact that LR-M was not a significant factor affecting the outcomes may suggest that this would not significantly affect the main outcomes of our study.

R2-4. The LRM is used instead of LR5 for analysis, but the LRM is used for malignant rather than HCC. This issue should be clarified in the study.

  • I hope the answer to this comment has been sufficiently made in the response to R2-3.

R2-5. In the figure of nomogram, the name of the group should be specified rather than 0 and1.

  • Perhaps you are referring to the nomogram point tables? If so, we would gladly change them to ‘no’ or ‘yes’, instead of 0 or 1.

R2-6. The patients were divided to 3 groups according to percentage of score<25, 25-75 and >75. The reason for this group needs to be clarified (1/4 to 4 groups or 1/3 to 3 groups).

  • The distribution of the nomogram points on the histogram showed a bell-shaped configuration. Therefore, it seemed reasonable to classify the patients based on quartiles. In addition, classifying the patients based on quartiles is a frequently used method. I hope the answer to your comment is sufficient.
